Medical Imaging with Deep Learning 2024

# Brain Artery Segmentation for Structural MRI

**Bertram Sabrowsky-Hirsch**[1]            BERTRAM.SABROWSKY-HIRSCH@RISC-SOFTWARE.AT
**Ahmed Alshenoudy**[1]                    AHMED.ALSHENOUDY@RISC-SOFTWARE.AT
**Stefan Thumfart**[1]                     STEFAN.THUMFART@RISC-SOFTWARE.AT
**Michael Giretzlehner**[1]                MICHAEL.GIRETZLEHNER@RISC-SOFTWARE.AT
[1] *Unit Medical Informatics, RISC Software GmbH, Hagenberg, Austria*

**Josef Scharinger**[2]                    JOSEF.SCHARINGER@JKU.AT
[2] *Johannes Kepler University, Linz, Austria*

## Abstract

The visualization of brain arteries in neuroimaging scans is essential for evaluating neurological disorders effectively. In this paper, we propose a deep learning-based method for the segmentation of brain arteries in structural MR images (sMRI), where their delineation poses a significant challenge due to the lack of contrast. Our fully automated strategy leverages two modules: one for generating pseudo labels from angiographic MR images and another for pairing these labels with five distinct sMRI sequences. The process enables us to construct a large dataset of 2626 labelled images for 669 patients used to train our segmentation model. In our experiments, our model achieved an average Dice Similarity Coefficient (DSC) of 0.66 across all sMRI around the central Circle of Willis structure in a 5-fold cross validation. We outline our results for each evaluated sMRI sequence, out of which we identify PD with a DSC of 0.7 as the best alternative to angiographic images.
**Keywords:** artificial intelligence, segmentation, brain arteries, sMRI

## 1. Introduction

Neurological disorders are often caused or directly intertwined with cerebrovascular diseases. If available, angiographic images such as the time-of-flight MR angiography (TOF) are generally preferable for the assessment of arteries due to their superior contrast and hence have been the subject of extensive research concerning automatic segmentation methods. However, the accurate delineation of arteries within structural MR images (sMRI) opens possibilities for a comprehensive assessment that includes the surrounding brain tissue. In that regard, works such as (Lee et al., 2019) have shown that sMRI are suitable for the assessment of brain arteries in the clinical practice. Furthermore, (Verleger et al., 2014) proposed a masked registration approach to significantly improve the alignment of brain arteries in TOF images. Thus, by segmenting arteries in sMRI we aim to similarly improve the alignment of sMRI and TOF images, primarily for accurate image fusion.

In our previous work (Sabrowsky-Hirsch et al., 2024b), we have shown that it is possible to automatically segment large brain arteries around the Circle of Willis (CoW) in sMRI and further annotate four circulatory regions in the arterial tree (left, right, anterior and posterior). However, this approach had the following limitations: It relied on automatically generated ground-truth annotations, which were limited to larger brain arteries, and was inhibited by the registration method (Klein et al., 2010) used to align the TOF and sMRI, which required meticulous supervision as it would frequently converge to a poor

alignment. We recently addressed the alignment problem through a robust geometric registration method (Sabrowsky-Hirsch et al., 2024a), in which we use annotated circulatory regions to compute accurate alignments.

In this paper, we propose an automated strategy to overcome the limitations of our previous approach: We use pseudo labels predicted based on gold-standard segmentations of brain arteries and improve the alignment of these labels with the sMRI through our geometry-based registration method. The process can be automatically applied to new data to retrain and improve the resulting segmentation model.

## 2. Methodology

Our experiments are based on a large collection of MR data outlined in Figure 1(a), which comprises four public datasets. Three are subsets of the IXI dataset[1] collected at three different institutions (GUYS, HH and IOP), and the TubeTK dataset[2] (TTK). Available MR sequences are TOF, PD, T1, T2, T1-Flash (FL) and T1-MPRage (MPR).

Figure 1(b) outlines our method, which combines two separate modules for segmentation and registration to prepare the final dataset $D$ used for training and evaluation of our model $M_C$. The segmentation module employs an nnU-Net (Isensee et al., 2021) model $M_A$ trained on publicly available gold-standard ground-truth for 45 patients of IXI (Chen et al., 2022) and 42 patients of TTK (Aylward and Bullitt, 2002), on which it achieved an average Dice Similarity Coefficient (DSC) of 0.82 in a 5-fold cross validation. The model was used to predict pseudo labels $L$ for all TOF images. The registration module employs our previous model for the annotation of large brain arteries $M_B$ (Sabrowsky-Hirsch et al., 2024b) to annotate the TOF and sMRI. The annotated vasculature was then aligned utilizing our geometry-based registration method (Sabrowsky-Hirsch et al., 2024a). The resulting transformations were finally used to warp and resample the sMRI to the space of the TOF images. In the final step, the TOF and the resampled $\overline{\text{sMRI}}$ together with the pseudo labels $L$ were used to train an nnU-Net model $M_C$ for the segmentation of brain arteries in all available MR sequences.

1. http://brain-development.org/ixidataset/
2. https://public.kitware.com/Wiki/TubeTK/Data

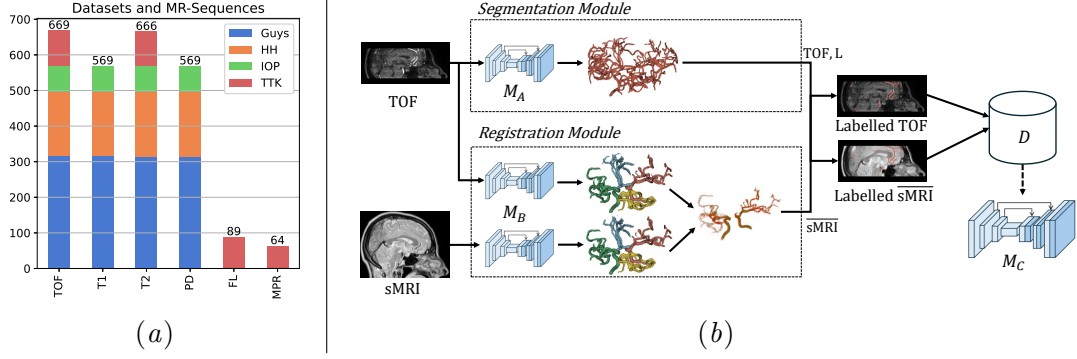

Figure 1: Overview of datasets and MR sequences (left) and our proposed method (right).

Table 1: DSC for each sMRI sequence, outlining different datasets and regions, respectively.

| Region | Dataset | TOF | PD | T1 | T2 | MPR | FL |
|---|---|---|---|---|---|---|---|
| Full | All | 0.91±0.06 | **0.57±0.06** | 0.49±0.05 | 0.53±0.06 | 0.46±0.06 | 0.40±0.05 |
| | Guys | 0.93±0.01 | **0.59±0.05** | 0.49±0.05 | 0.55±0.05 | - | - |
| | HH | 0.94±0.01 | 0.56±0.06 | 0.52±0.04 | 0.51±0.06 | - | - |
| | IOP | 0.86±0.04 | 0.55±0.04 | 0.43±0.04 | 0.53±0.05 | - | - |
| | TTK | 0.82±0.08 | - | - | 0.48±0.06 | 0.46±0.06 | 0.40±0.05 |
| CoW | All | 0.94±0.04 | **0.70±0.06** | 0.64±0.06 | 0.67±0.07 | 0.61±0.08 | 0.58±0.06 |
| | Guys | 0.95±0.01 | **0.71±0.07** | 0.64±0.07 | 0.69±0.07 | - | - |
| | HH | 0.96±0.01 | 0.69±0.06 | 0.67±0.04 | 0.66±0.06 | - | - |
| | IOP | 0.91±0.02 | 0.70±0.04 | 0.61±0.05 | 0.68±0.04 | - | - |
| | TTK | 0.87±0.07 | - | - | 0.60±0.07 | 0.61±0.08 | 0.58±0.06 |

## 3. Results and Discussion

We trained and evaluated the model in a 5-fold cross validation and report results in Table 1 and Figure 2. The DSC metric was evaluated in both, the full image region and a smaller region around the CoW, respectively. Most of the deviation from the ground-truth can be attributed to undersegmentation of smaller arteries in the sMRI, which is strongest outside the CoW region. Larger vessels are accurately detected across all sMRI. This observation can be partially explained by the generally lower resolution of the sMRI, which inhibits the detection of smaller vessels. The PD sequence yielded the best results followed by T2.

**Conclusion:** We proposed a novel method for the segmentation of brain arteries in sMRI, which extends our previous work on the topic. Using our method, new data can be automatically processed to extend the training dataset and further improve the model. In our next steps we plan to further refine the alignment of arteries through a non-linear registration step. Details on our registration method as well as an upcoming in-depth publication will be shared with our evaluation code at: github.com/risc-mi/cerebral-artery-segmentation.

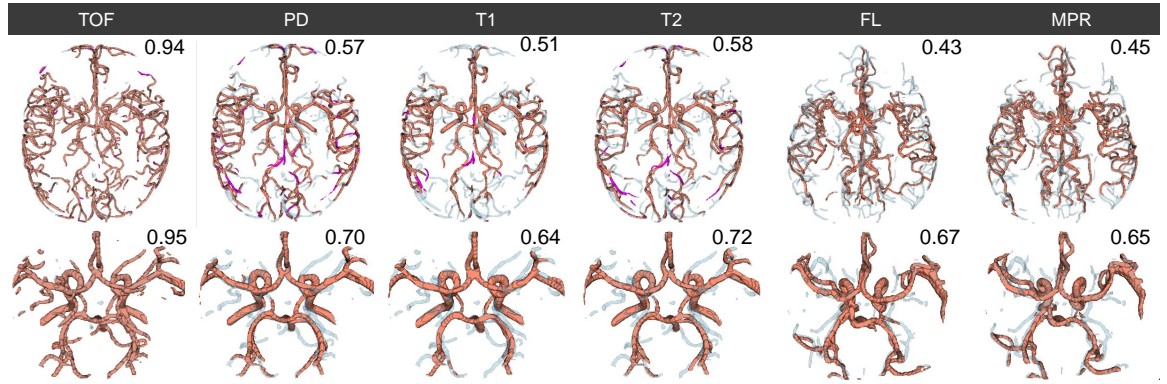

Figure 2: Visualizations and DSC scores for the full (top) and CoW region (bottom) of examples for each MR sequence. Oversegmentation is shown in magenta and undersegmented ground-truth in light-blue.

## Acknowledgments

This work was funded by the FFG (Austrian Research Promotion Agency) under the grant 872604 (MEDUSA) and research subsidies granted by the government of Upper Austria. RISC Software GmbH is a member of UAR (Upper Austrian Research) Innovation Network.

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
