# OpenReview forum: "Brain Artery Segmentation for Structural MRI"
_MIDL.io/2024/Short_Papers — MIDL 2024 Short Papers_

### Official Review · Reviewer_L3tr · 2024-04-23

**Confidence:** 5
**Final Rating:** 4

**Review:**

The paper describes an approach to the segmentation of blood vessels in structural MRI, a challenging problem. Authors leverage information from TOF to obtain a (weak) ground-truth segmentation mask in structural MRI images using registration. The combination of segmentation, registration, and a clean experimental set-up is of interest to the MIDL community, as are the results obtained.

Strengths
- Well-written abstract, easy to follow.
- Results are comparable across different cohorts.
- Authors provide code.

Weaknesses
- There are no ground truth annotations in structural MRI with which the authors can compare. Instead, all evaluation is done w.r.t. automatic segmentations that have been propagated from TOF to structural MRI.
- Authors compare different structural image sequences (e.g., PD vs. T1 vs T2), but do not show results on the combination of these. It would be interesting to see if performance can be improved by using, e.g., PD + T1 + T2 as inputs to a segmentation model. Results in Fig. 2 indicate that slightly different errors are made based on

---

### Decision · Program_Chairs · 2024-04-26

Accept